# Physiological Performance of *Pyrus pyraster* L. (Burgsd.) and *Sorbus torminalis* (L.) Crantz Seedlings under Drought Treatment

**DOI:** 10.3390/plants9111496

**Published:** 2020-11-05

**Authors:** Viera Paganová, Marek Hus, Zuzana Jureková

**Affiliations:** 1Faculty of Horticulture and Landscape Engineering, Slovak University of Agriculture, 949 76 Nitra, Slovakia; xhus@uniag.sk; 2Faculty of European Studies and Regional Development, Slovak University of Agriculture, 949 76 Nitra, Slovakia; zuzana.jurekova@uniag.sk

**Keywords:** water regime, leaf gas exchange, stomatal conductance, growth, biomass

## Abstract

In this study, seedlings of *Pyrus pyraster* and *Sorbus torminalis* were grown for 60 days in the regulated environment of a growth chamber under different water regimes. The measured indicators were the growth and distribution of mass to organs, total biomass, root to shoot mass ratio (R:S), and gas exchange parameters (g_s_, E, A_n_, and water use efficiency (WUE)). The amount of total biomass was negatively affected by drought. Differences between species were confirmed only for the dry matter of the leaves. *P. pyraster* maintained the ratio of the mass distribution between belowground and aboveground organs in both variants of the water regime. *S. torminalis* created more root length for a given dry-mass under drought treatment, but its R:S was lower compared to *P. pyraster*. The water potential of the leaves (Ψ_wl_) was affected by substrate saturation and interspecific differences. *P. pyraster* had a demonstrably higher water potential and maintained this difference even after prolonged exposure to drought. After 30 days of different water regimes, *Pyrus* maintained higher values of g_s_, A_n_, and E in control and drought treatments, but over a longer period of drought (after 50 days), the differences between species were equalized. The changes of the leaf gas exchange for *Pyrus* were accompanied by a significant increase in WUE, which was most pronounced on the 40th day of the experiment. A significant and strong relationship between WUE and g_s_ was demonstrated. The results confirmed the different physiological performances of seedlings of tree species and the different mechanisms of their response to water scarcity during drought treatment. *P. pyraster* presented more acclimation traits, which allowed this taxon to exhibit better performance over a longer period of water scarcity.

## 1. Introduction

The increasing frequency of drought events in Central Europe has adverse effects on tree life, reducing the biodiversity and ecological value of the ecosystems. In particular, tree seedlings with a shallow or undeveloped root system suffer and often die from drought [1,2,3]. Therefore, species with higher resistance to drought, which maintain higher productivity and intensive photosynthesis even under a lack of water, receive attention especially in locations which are prone to drought. The identification of such species is important as climate change is expected to increase the number of drought-prone areas worldwide [4,5].

Drought reduces plant productivity [6]; therefore, measurable indicators need to be defined as well as quantified to reveal the effects of drought on plants. Drought-tolerant trees are able to withstand short-term fluctuations in water supply as a result of their physiological and metabolic functions. According to Kunz et al. [7], resistance is the ability of plants to withstand stress and can be quantified as the ratio between their physiological performance during drought and under normal conditions without drought stress. Gregory et al. [8] and Gebrekirstos et al. [9] characterize resistance by the time course of three physiological processes: the rate of photosynthesis (An), rate of transpiration (E), and stomatal conductivity (g_s_).

Klein [10] considers stomatal conductivity (g_s_) and leaf water potential (Ψ_wl_) to be key characteristics for understanding plant functions under changing climatic conditions. The experimental and theoretical basis of the stomatal conductivity (g_s_) model and indicators of the leaf and stand water regime were provided by Tardieau and Davies, [11]. Hammer et al. [12] consider the Tardieau–Davies model suitable for the evaluation of the mentioned relationships and their consequences in short-term and long-term intervals in a given climate scenario.

Water use efficiency (WUE) is used to analyze the influence of environmental factors and structural and functional traits of plants, as well as to optimize the relationship between dry matter production and water consumption. It is the ratio of the rate of photosynthesis and rate of transpiration (A_n_/E) and refers to the gas exchange of assimilation organs over a short period of time. The WUE for carbon assimilation needs to be studied at the leaf level along with responses to physical environmental factors. Hatfield and Dold [13] predict that WUE will increase as a result of climate change. According to their data, it will be necessary to identify genotypes that are capable of intensive carbon assimilation even under conditions of water stress. To understand the impact of climate change on WUE, we need to determine the impact of drought on the growth and use of water under controlled conditions. Furthermore, it is important to determine the interactions of the physical and biological factors of genotypes that are able to use water more efficiently without significant costs for photosynthesis.

Knowledge of the parameters of physiological performance of young trees is important for further study of the physiological mechanisms regulating utilization and management of water. It is important also for the economic analysis of the tree ability survive/withstand drought events. The quantitative description of the relationships between the physiological processes and environmental factors provides the background for creation of the ecological models describing acclimation and adaptive behavior of plants [14]. These models and simulations composed of the ecophysiological measurements will also be used for prognosis and planning of the natural ecosystem’s recovery after disturbance.

In the natural conditions, the study of woody plants responses to extreme climatic events (drought) can be difficult, due to the large species, ontogenetic and age heterogeneity of their communities. The field measurements of the eco-physiological parameters should be supplemented by experiments held under regulated and controlled conditions, where is possible to obtain reproducible data and identify the limit values of the investigated factors.

Changing environmental conditions have significant impact on the growth and survival of woody plants in both, natural and urban environments. The identification of adaptable species that are able survive under longer periods of drought is essential for sustainability of the biomass production and environmental benefits of woody plants in the context of climate change. The subjects of the presented research are tree species, *Pyrus pyraster* L. (Burgsd.) and *Sorbus torminalis* (L.) Crantz, which belong to the natural European Flora; both studied species are considered to be light-demanding woody plants [15,16,17,18,19,20] which can grow even on sites with periodic drought events [15,18,19,20].

*S. torminalis* is considered to be a submediterranean species [21,22]. It requires a warm climate specific to oak-dominated forests (*Quercetalia pubescentis*, *Quercetalia robori-petraeae*, *Querco robori-Carpinenion betuli)* and beech forests on calcareous stands (*Cephalanthero-Fagenion*). *S. torminalis* prefers rich, deep, and fresh soils which are continuously supplied with water [19,23]; however, it also grows in locations influenced by a short-term water deficit in soils [20]. According to the findings of Wilhelm [24], *S. torminalis* is more tolerant to shading than *P. pyraster*.

*P. pyraster* is the native pear species in central Europe distributed across a large area within the temperate zone [25,26]. The wild *Pyrus* species grow also in more xeric Mediterranean or steppe environment [27]. Due to the high light demands, *P. pyraster* occurs in rather extreme or marginal site conditions, where competition with other tree species is weakened [28]. This taxon grows on almost all soil types, except for extremely acidic soil. Its quite deep tap root system permits successful growth on very dry soils [18]. According to the Ellenberg’s moisture scheme *P. pyraster* has quite wide ecological amplitude [29]. This taxon is tolerant to short-term flooding and appears in the communities of hardwood floodplain forests (*Ulmeto-Quercetum*) [30,31].

The purpose of the presented study was to determine the physiological performance of the above-ground and underground organs of two juvenile tree species *P. pyraster* and *S. torminalis* in conditions of water scarcity. We assume that it is a local stress affecting the plant roots. Therefore, it was of interest to investigate (1) how will change the physiological performance of the aboveground organs (parameters g_s_, A_n_, E, RWC and Ψ_wl_) of the studied taxa in relation to the documented interspecies variability; (2) whether expected differences in the physiological performance of the studied species will have impact on their growth and biomass production; (3) whether obtained findings can be applied for selection of the tree species more resilient to climate change?

The aims of study were (1) to quantify the physiological performance of the juvenile plants of *P. pyraster* and *S. torminalis* under drought conditions; (2) to determine the effect of drought events on seedling growth, parameters of photosynthesis and transpiration of the studied taxa; (3) to verify the relationship between Ψ_wl_, A_n_, E and g_s_ of the leaves of experimental plants.

## 2. Material and Methods

### 2.1. Plant Material

*S. tominalis* inhabits mostly horizontal terrains or sun-facing (south, south-west, and south-east) slopes [19,32,33]. The majority of sites with *S. torminalis* have been found at lower altitudes of 200–450 m. The majority of stands with *P. pyraster* have been found at altitudes up to 500 m [18]; however, *P. pyraster* appears also in cold climates in the mountain areas and can be found at altitudes up to 1400 m.

The plants used in the experiment were grown from seeds collected from locations in Slovakia and in Czech Republic; the characteristics are given in Table 1. The selected provenances represent optimum growth conditions of the studied taxa within their natural area of distribution in Central Europe; the climatic conditions of their stands are rather similar. The seeds were extracted manually after harvest and subjected to a cold stratification treatment outdoors for 90 days with temperatures ranging from −10 °C to +5 °C.

Seeds were germinated in plastic plates filled with the peat-based sowing substrate (pH of 5.5–6.5, enriched with nutrients 0.5 kg/m^3^ NPK ratio 12:14:24) in a cold greenhouse under natural light conditions with temperatures ranging between 5–15 °C. The growth rates of *P. pyraster* and *S. torminalis* juvenile plants are different; therefore, plants of both taxa were selected for experiment at the time when they reached principal growth stage 1: leaf development, after the cotyledons have been completely unfolded (BBCH10) according to the BBCH-scale [35].

During the phenological growth stage ”cotyledons completely unfolded”, the seedlings were placed in plastic pots (90 mm in diameter, volume 0.47 L) with a fertilized peat-based growth substrate (20% black peat and 80% white peat moss, 0–5 mm fraction, pH of 5.5–6.5, enriched with nutrients 1.0 kg/m^3^ NPK 14:16:18). Each pot was placed in the plastic bag to avoid uncontrolled water leakage.

At the beginning of the experiment, the plant biomass per unit rooting volume was calculated for both species to avoid the risk of pot size having an effect on growth. According to the findings of Poorter et al [36], the plant biomass to pot volume should not be larger than 2 g·L^−1^; thus, the plant biomass to pot volume ratio was calculated for *P. pyraster* (0.89 g·L^−1^), as well as for *S. torminalis* (1.06 g·L^−1^).

### 2.2. Experimental Design

The experimental plants were cultivated in different water regimes, where drought was considered as 40% water according to the weight of the fully saturated substrate, and the control condition was considered as 80% water according to the weight of the fully saturated substrate. The water content in the growth substrate was calculated based on wet weight [37].
$$M_{n}=\frac{\left(W_{w}-W_{d}\right)}{W_{w}}\times 100$$
M_n_ = moisture content (%) of material nW_w_ = wet weight of the sampleW_d_ = weight of the sample after drying

The different water regimes (drought and control) were maintained by regularly weighing the pots on a precision industrial scale (Kern & Sohn GmbH, Balingen Germany) with laboratory accuracy (max = 8000 g, standard deviation = 0.05 g) at 2-day intervals. There were 12 replications for each substrate saturation level.

After 14 days of acclimatization (Figure 1), the plants were maintained under different water regimes for 60 days from May to July in the growth chamber PolEko KK1450. The photoperiod of the growth chamber was set to 14/10 h; the irradiation density on the surface of the uppermost leaves was 202.5 μmol m^−2^·s^−1^. Air humidity was 65%, and the temperature was maintained at 22 °C during the light period and 14 °C during the dark period. The plants were randomly placed and rotated within the chamber once a week in order to avoid a potential within-chamber effect [38,39].

Measurements of leaf gas exchange were carried out at the 30th, 40th, and 50th day of the experiment. Water potential measurements were performed at the 40th and 50th day of the experiment. Earlier measurements were not feasible due the small size of the seedlings. The measurements of morphometric traits, as well as the determination of the biomass distribution in the plant organs, were performed at the end of the experiment.

### 2.3. Measurement and Analysis of Plant Parameters

The total fresh mass of all individuals was determined before seedlings were planted in the pots and also at the end of the experiment. Before weighing, the plant roots were gently extracted from the growth substrate by hand and carefully washed to minimize fine root loss.

The WinRhizo REG 2009 system (Regent Instruments, Québec, QC, Canada, SK0410192) was used for the measurement of the root length (mm). The length of the primary stem of the experimental plants was also measured, and the total leaf area (LA) was determined by scanning fresh leaves using ImageJ software.

The dry weight of the plant organs was determined after the plant material was dried at 105 °C until it reached a constant weight. Other parameters calculated were the leaf water content (LWC), the specific root length (SRL), the specific leaf area (SLA), and the root to shoot ratio (R:S). SLA was calculated as the ratio of leaf area to leaf dry mass [40].

### 2.4. Leaf Gas Exchange

The net photosynthetic rate (A_n_), stomatal conductance (g_s_), transpiration rate (E), and water use efficiency (WUE) were measured, beginning 30 days after the initiation of the different water regimes and then twice at 10-day intervals. The measurements were performed using the gasometer CIRAS-3 (PP-systems, Amesbury, MA, USA) attached to a PLC3 universal leaf cuvette fitted with a 1.75 cm^2^ measurement window, on the fully expanded leaf for each plant on the upper part of the seedling. The determination of leaf gas exchange was performed between 8 a.m. and 11 a.m. The molar flow rate of air entering the leaf chamber was kept constant at 300 cm^3^·min^−1^. The average leaf temperature was maintained near 26 °C (±0.26 °C SD), the vapor partial pressure deficit was 1.38 ± 0.25 kPa, the photosynthetically active radiation (PAR) was kept constant at 250 μmol·m^−2^·s^−1^, and the CO_2_ concentration was kept constant at 400 µmol·mol^−1^. Upon clumping the leaf in the cuvette, measurements were taken after the full stabilization of A_n_ and g_s_, which took up to 5 min. The actual measurement of leaf gas exchange lasted for 5 min per seedling.

### 2.5. Leaf Water Potential and Relative Water Content

The water potential of the leaf tissues (Ψ_wl_) was determined by psychrometric measurement performed by Wescor (model Psypro, EliTech Incc, Logan, UT, USA) using a C-52 sample chamber at an ambient temperature of 21 °C. The water potential of leaf tissue was measured on the fifth leaf in the central part of the experimental plants. The leaf samples were taken from three plants of each taxon in each of the two variants of water regime. The measurements were performed at the 40th and 50th day of experiment from 7 a.m. to 4 p.m. in three replicates for each taxon and water regime. Data were analyzed by a multifactor analysis of variance (ANOVA) to detect significant factors (water regime/taxon) influencing water potential.

Relative water content (RWC; %) was determined by gravimetric method according to Barrs and Weatherley [41] with 4 h saturation of leaf samples in water at 4 °C in the dark. RWC was calculated as: RWC = [(FW − DW)/(SW − DW)] × 100, where FW is fresh weight, DW is dry weight and SW is the weight after full saturation of leaf samples.

### 2.6. Statistical Analysis

Mathematical and statistical data analysis was performed using the Statgraphics Centurion XVII software (StatPoint Technologies, Warrenton, VA, USA, XVIII, license number: B480-E10A-00EA-P00S-60PO).

Analyses of normality and homogeneity of variance for all variables were performed with Shapiro–Wilk’s test (at significance level of α = 0.001) and Leven’s test (at significance level of α = 0.05). Grubbs’ test was used to detect and remove single outliers in the experimental data set. Variables that did not fulfil the requirements for an analysis of variance were log transformed.

The parameters of the plant organs were analyzed with a two-way analysis of variance (ANOVA) assuming the level of growth of substrate saturation (drought and control) and tree species as fixed effects. Differences within treatments and tree species were subsequently tested with the Tukey honest significant difference (HSD) test.

The repeated measurements of the leaf gas exchange and water potential of the leaf tissues were conducted periodically on the 30th, 40th, and 50th day of drought treatment. The leaf gas exchange and water potential of leaf tissues were analyzed using a multifactor analysis of variance with the species, treatment, and duration of experiment as fixed factors. Differences within treatments, tree species, and measurements were subsequently tested with the Tukey HSD test at significance levels of 0.05 and 0.01.

## 3. Results

### 3.1. Growth and Biomass Allocation in the Plant Organs

In the juvenile stage of growth, the studied tree species differed in terms of the mass formation and allocation in the plant organs (Table 2). Compared to *P. pyraster*, *S. torminalis* had larger leaf area, and higher values of root length and specific root length. It maintained higher values of SRL and LA parameters even during drought treatment. *P. pyraster* had significantly higher stem increment and stem length (Figure 2 and Figure 3) and invested more to root growth (higher R:S), even under drought treatment. Under a lack of water, *P. pyraster* formed thicker leaves (SLA = 16.45 mm^2^·mg^−1^).

The biomass formation was negatively affected by drought in both species: *P. pyraster* (−44%), *S. torminalis* (−54%). The interspecific differences were not confirmed for the total biomass (Table 2), only for the dry weight of leaves (DW_L_). *S. torminalis* accumulated more dry mass in the leaves in control conditions (+56%) as well as under drought treatment (+38%). *P. pyraster*, which has high demands on light, presented intensive stem growth already in the juvenile stage of growth and, compared to *S. torminalis*, allocated more dry mass in the root (higher R:S values). *S. torminalis* created more root length for a given dry-mass investment (SRL) in control conditions, as well as under drought treatment (Table 2).

### 3.2. Leaf Gas Exchange

Significant differences between *P. pyraster* and *S. torminalis* were identified in all leaf gas exchange parameters based on the results of the multifactor ANOVA. We also tested the impacts of the treatment (regime) and duration of experiment (measurement) as fixed factors. The null hypothesis was adopted for A_n_ under different treatments (regimes). The duration of the experiment with the drought treatment of the seedlings had a significant impact on all studied leaf gas exchange parameters except A_n_. Quantitative data analyses were performed, including multiple comparisons of means using Tukey’s HSD procedure (Table 3).

Significant differences in the physiological performance of the studied species were found after 30 days of the different water regimes. Compared to *S. torminalis*, *P. pyraster* had higher values of g_s_, A_n_, and E under control and drought treatments (Table 3). The most significant differences between the studied species were manifested in g_s_. The parameter was four-times higher (control) and five-times higher (drought) for *P. pyraster* seedlings compared to *S. torminalis*. After 40 days of the different water regimes, significant interspecific differences were found in the parameters g_s_, A_n_, and E only under drought treatment. After 50 days of the different water regimes, demonstrable interspecific differences were observed only for control plants for the parameters g_s_ and E (Table 3).

A more detailed analysis of the gas exchange parameters based on the duration of the different water regimes shows that *S. torminalis* maintained relatively low values of g_s_ in both regimes (control and drought) without significant differences between measurements (Figure 4a). During the experiment, the balanced physiological performance of *S. torminalis* was shown under control, along with balanced values of A_n_ and E (Figure 4b,c). These parameters were not affected by the different water regime (Table 3). The duration of the drought treatment affected A_n_ (Figure 4b). The differences were noticed also for the WUE of the control and drought-treated seedlings, with a significant increase in the mean values on the 40th day of the experiment (Figure 4d). The mean values of WUE were 12.93 ± 2.97 mmol CO_2_ mol^−1^ H_2_O (control) and 17.13 ± 5.59 mmol CO_2_ mol^−1^ H_2_O (drought).

*P. pyraster* maintained a balanced course of A_n_ in the control and also under drought treatment for 40 days. A demonstrable decrease of the A_n_ was not recorded in the drought treatment until the third measurement (50th day) (Figure 4b). These changes in A_n_ and E were accompanied by a significant increase in values of WUE for *P. pyraster* (Figure 4d). After 30 days of the drought treatment, the mean value of WUE was 4.60 ± 0.48 mmol CO_2_·mol^−1^ H_2_O. After 40 days, the mean value increased significantly to 13.02 ± 3.32 mmol CO_2_·mol^−1^ H_2_O.

A strong correlation was found between the WUE and g_s_ parameters for *P. pyraster* (r = 0.914639, *p* < 0.05) (Figure 5a). The relationship between the above-mentioned physiological parameters is highly significant and closely correlated. The values of WUE for *P. pyraster* significantly increased following the reduction of g_s_ to under 80 mmol H_2_O m^−2^·s^−1^. In contrast, the results of the regression analysis for *S. torminalis* show a weak correlation relationship between WUE and g_s_ (r = −0.568677, *p* < 0.05) (Figure 5b). The seedlings maintained lower values of g_s_ (g_s_ < 60 mmol H_2_O m^−2^·s^−1^), and the changes of WUE were insignificant.

Interspecific differences were recorded also in the values of A_n_. In the different water regimes, the A_n_ values for *P. pyraster* increased sharply following the increase of g_s_ to 60 mmol H_2_O m^−2^·s^−1^; then, values of A_n_ oscillated within a relatively wide range even at higher values of g_s_ (Figure 6a). In contrast, the rise of the A_n_ values was linearly correlated with g_s_ values for *S. torminalis* (Figure 6b).

### 3.3. Leaf Water Status

The water potential of experimental plants was significantly influenced by the taxon and level of the growth substrate saturation (Table 4). The water potential of leaf tissues was lower in *S. torminalis* compared with *P. pyraster.* In the control, Ψ_wl_ was −0.81 MPa and −0.84 MPa. After 40 days of drought treatment, Ψ_wl_ decreased to −1.15 MPa, and after 50 days of drought treatment, this dropped to −1.53 MPa. The water potential of *P. pyraster* leaf tissues was also affected by drought. After 40 days of drought treatment, the value of Ψ_wl_ decreased to −1.27 MPa, and after 50 days of drought treatment, this decreased to −1.33 MPa. RWC values for control plants demonstrated interspecific differences after 50 days of the experiment (Table 4), when *S. torminalis* had significantly lower RWC (89.30%) compared to *P. pyraster* (94.56%). RWC values for drought-treated plants decreased significantly in both species (RWC˂ 80%), but interspecific differences were not confirmed.

The obtained data describe the different properties of the studied taxa under conditions of different water regimes (control and drought) and indicate the potential capacity of plants to take up the available water. *P. pyraster* had significantly higher water potential of the leaf tissues in comparison with *S. torminalis* and maintained this difference even after prolonged exposure to drought (50 days).

## 4. Discussion

The processes of photosynthesis and transpiration carried out in the plant leaves are reflected in the relationship between plant productivity and water use. Water scarcity is a crucial factor that negatively affects this relationship because it is limiting for growth and the utilization of water. Lack of water significantly inhibits CO_2_ exchange and the rate of transpiration, and both processes are controlled by the stomatal conductance (g_s_). According to [42], stomatal conductance is the evidence of physiological performance because it expresses the rate of CO_2_ availability for photosynthesis (A_n_) and the management of water.

The impact of drought on the physiological performance of *P. pyraster* and *S. torminalis* seedlings was studied in our measurements, and we found significant interspecific differences. *P. pyraster* had higher values of all measured parameters both in the control and under drought treatment. The most marked differences were found in g_s_. The values varied significantly between the species; compared with *Sorbus,* the values were several times higher for *Pyrus*. This parameter changed against the background of the changes in leaf water potential, which decreased after 40 days of drought treatment to −1.27 MPa and after 50 days to −1.33 MPa. The stomatal conductance decreased in both drought and control conditions, depending on the duration of exposure to the different water regimes (control and drought). According to [10,43], the stomatal conductance and the leaf water potential interact on the principle of feedback, which results to stomatal sensitivity, i.e., a decrease in g_s_ occurs in response to the decreasing leaf water potential. In *Pyrus* leaves, transpiration (E) was reduced as well, according to the decrease of g_s_ in both regimes of the substrate saturation by water. After 40 days of the experiment, balanced A_n_ values were maintained despite the decreasing g_s_ for both control and drought-treated plants. The factor of water shortage did not show a negative impact on photosynthesis until the 50th day of drought treatment, when a significant decrease of A_n_ was found.

A lower water potential was found in the leaves of *S. torminalis* seedlings. It decreased over the entire duration of the experiment from −0.81 MPa to −1.53 MPa and was accompanied by low and balanced g_s_ values with non-significant differences between measurements. According to [44], g_s_ reduces the water potential of leaves and negatively affects the leaf expansion. On the other hand, it has a positive effect on the photosynthesis rate. This was not confirmed in our measurements, because *S. torminalis* showed low photosynthesis and transpiration rates and formed a larger leaf area.

When comparing the characteristics of the studied species, *S. torminalis* had relatively low g_s_ values in both regimes of irrigation, and the decrease of g_s_ due to stress was insignificant, regardless of the duration of the drought treatment. After 50 days of the different water regimes, the g_s_ of the drought-treated seedlings were equal to the control. The other measured parameters, A_n_ and E, showed balanced and relatively low values in the control as well as during drought treatment. We did not confirm the data of [7,45], who found rapid changes in the physiological performance of *S. torminalis* after 32 days of drought treatment. The authors explain these changes by the rapid decrease of g_s_ recorded after the restricted availability of water. Zhang et al. [42] confirmed a nonsignificant correlation between g_s_ and WUE, but found a significant correlation between A_n_ and E. In our experiment, *S. torminalis* maintained low values of g_s_, and we confirmed a weak correlation between g_s_ and WUE (r = −0.41, *p* <0.05).

There were not confirmed interspecific differences in total biomass production, but *S torminalis* despite the low physiological performance distributed more dry mass to the leaves in control (+56%) and in drought (+38%) conditions. The seedlings also created more root length for a given dry mass investment, which was reflected in higher SRL values.

We accept the opinion of [46], whose hypothesis was based on the assumption that plant productivity (maize in their experiment) can increase without changing the amount of water used; i.e., with an unchanged WUE. Caldeira et al. [44] suggest that particularly expansive volume growth is negatively correlated with g_s_ and daytime transpiration rate, which is typical for the genotypes with the lowest g_s_ values.

The interspecific differences are also interesting due to the difference found in WUE values. The changes in A_n_ and E were accompanied by a significant increase in the water use of *P. pyraster* seedlings, and regression analysis confirmed that WUE demonstrably increased with decreasing g_s_. *S. torminalis* had low A_n_ and E values, leading to a decrease in WUE values. Kunz et al. [7], in a study of the effects of drought on the growth, rewetting, and gas exchange of several European deciduous trees, concluded that *S. torminalis* is a less drought-resistant species compared to *Acer campestre* and *Acer platanoides*, as its physiological performance declined rapidly when the seedlings were affected by drought. In accordance with this opinion, and based on our measurements, it can be added that seedlings under conditions of water deficiency reduce the exchange of CO_2_ and H_2_O to prevent drought damage. The strategy used by *S. torminalis* is to use and spend available water. Zhang et al. [42] also investigated the effect of drought on the leaf gas exchange of various tree species, herbs, and climber plants. The authors found a significant inhibition of CO_2_ exchange in the leaves of the deciduous tree species induced by drought. The inhibitory effect escalated depending on the duration of stress. Drought significantly reduced A_n_, E, and g_s_; however, this occurred according to the characteristics of the species.

Within our study a moderately strong relationship has been confirmed between A_n_ and g_s_ for *P. pyraster* and *S. torminalis*. In agreement with findings of Héroult et al. [47] we confirmed, that under controlled conditions, species with different ecological amplitude to the soil moisture demonstrated different magnitude in response to water regime. According to several studies, the differences in stomatal conductance and its relationship with photosynthesis for humid zone species compared with drought-tolerant species, suggest strong selection for leaf and whole-plant characteristics and different stomatal optimization behavior for species from climates with contrasting rainfall and drought frequencies [47,48,49]. In our study, significant differences have been confirmed for leaf and root characteristics, where *P. pyraster* created thicker leaves (SLA) and *S. torminalis* invested more dry mass to root length (SRL) under conditions of water scarcity.

*Pyrus* maintains it physiological performance even under conditions of increased and prolonged water shortage. We observed decreased g_s_ and E values; however, a higher photosynthetic rate was maintained for a longer period of time. The significant decrease of A_n_ occurred after 50 days of drought treatment. Dias and Brüggemann [50] considered a significant increase of WUE (on the 40th day of experiment) accompanied by decreased or low values of A_n_ and E to be a common reaction of various tree species to the reduction of soil moisture. On the other hand, the increase of WUE together with the decrease in functions related to CO_2_ and H_2_O exchange is considered to be an indicator of higher drought resistance. Under drought, *Pyrus* significantly increases the efficiency of water use; this strategy is used to save water.

## 5. Conclusions

Interspecific differences were identified in the physiological performance of *P. pyraster* and *S. torminalis* seedlings in drought in this experimental research work. It was confirmed that the gain of CO_2_ and the loss of water by transpiration are species-dependent and significantly influenced by stomatal conductance (g_s_). This variable endogenous factor exhibited several times higher values in *P. pyraster* leaves compared with *S. torminalis* in the control, as well as for plants under drought treatment. *Sorbus* maintained a low g_s_ during the experiment in both control and drought conditions.

Under conditions of water shortage, the strategy of the studied tree species was based not only on the ability to assimilate C and the transpiration of water, but also on the mass balance for a specific period of time. The results show that *P. pyraster* maintained the mass distribution ratio between belowground and aboveground organs, even in conditions of water scarcity, and the R:S ratio did not change significantly. In drought, *Pyrus* formed thicker leaves (lower SLA) and reduced stomatal conductance and transpiration; additionally, it maintained a higher rate of photosynthesis. *Sorbus* created more root length for a given dry-mass investment (SRL), although the R:S ratio was lower compared to *Pyrus*. The seedlings reduced the assimilation of CO_2_ and H_2_O transpiration to prevent drought damage. In response to drought, *S. torminalis* applied a water-saving strategy, maintaining low leaf stomatal conductance. The changes of the leaf gas exchange for *Pyrus* were accompanied by a significant increase in WUE, which was most pronounced on the 40th day of the drought treatment. We have confirmed a significant and strong relationship between WUE and g_s_ for *Pyrus.* The correlation between WUE and g_s_ for *Sorbus* was weak.

Our results confirmed the different physiological performances of seedlings of tree species and the use of different mechanisms in their response to water scarcity and duration of drought.

## Figures and Tables

**Figure 1 plants-09-01496-f001:**
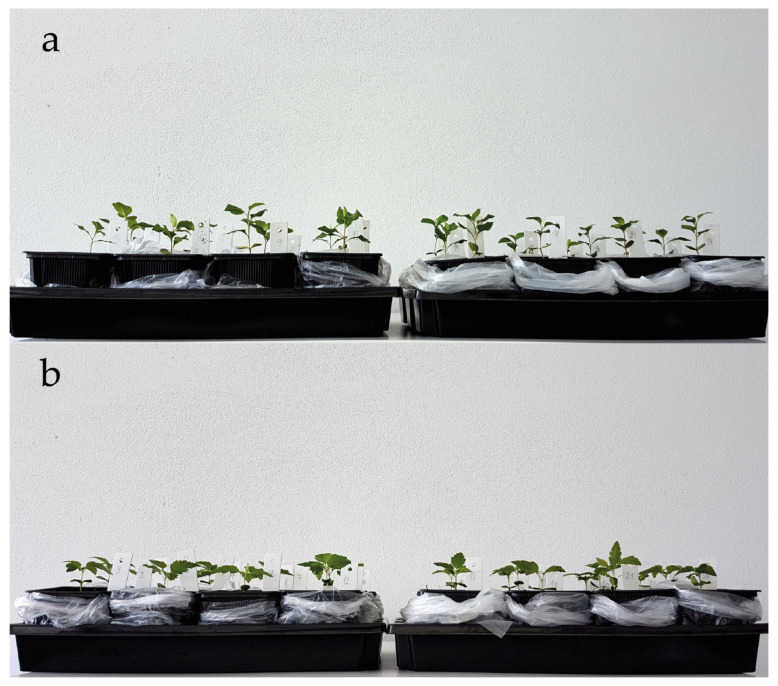
The seedlings of (**a**) *P. pyraster* and (**b**) *S. torminalis* after acclimatization in the beginning of experiment with different water regime; (**left**) control plants; (**right**) plants grown under drought treatment.

**Figure 2 plants-09-01496-f002:**
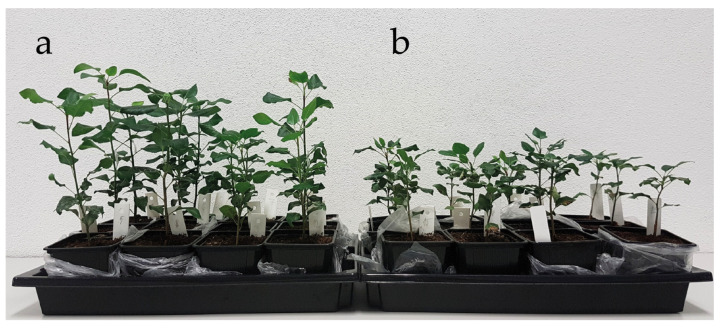
The differences in growth and biomass formation of *P.pyraster* seedlings grown for 50 days in different water regimes; (**a**) control plants; (**b**) plants under drought treatment.

**Figure 3 plants-09-01496-f003:**
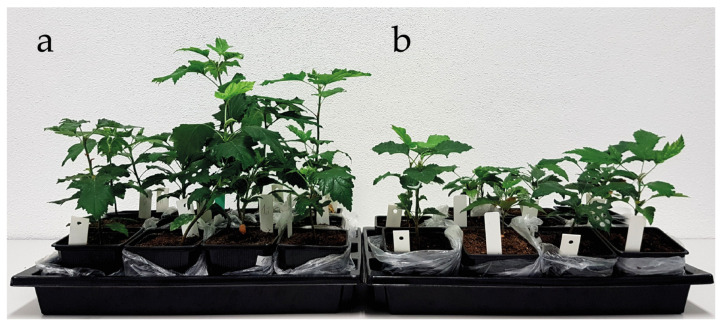
The differences in growth and biomass formation of *S. torminalis* seedlings grown for 50 days in different water regimes for 50 days; (**a**) control plants; (**b**) plants under drought treatment.

**Figure 4 plants-09-01496-f004:**
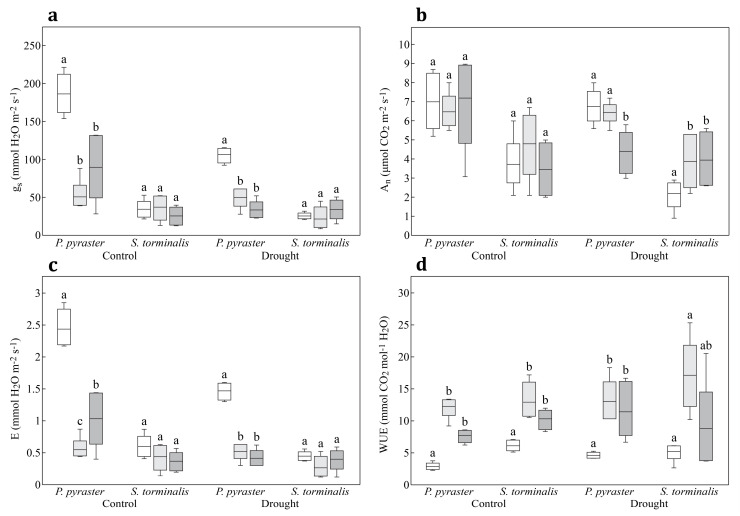
Box plots for (**a**) stomatal conductance (g_s_); (**b**) net photosynthetic rate (A_n_); (**c**) transpiration rate (E); and (**d**) water use efficiency (WUE) for seedlings of *P. pyraster* and *S. torminalis* measured after 30 days (white bars), 40 days (light grey bars), and 50 days (dark grey bars) of different water regimes. Drought was considered as 40% water according to the weight of the fully saturated substrate and control as 80% water according to the weight of the fully saturated substrate. The significant differences affected by the duration of drought treatment (p < 0.05) are denoted by different letters.

**Figure 5 plants-09-01496-f005:**
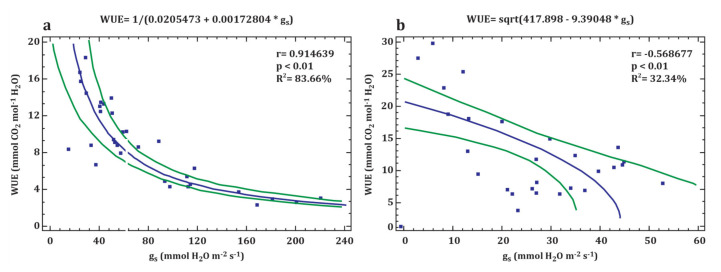
Relationship between the water use efficiency (WUE) and stomatal conductance (g_s_) of *P. pyraster* (**a**) and *S. torminalis* (**b**) seedlings grown for 50 days under different water regimes. The plot shows regression curve (blue line) and 95% confidence intervals (green lines). There were species specific responses to drought—a strong relationship was confirmed between WUE and g_s_ for *P. pyraster* (**5a**). During the experiment, *S. torminalis* maintained quite low and balanced values of g_s_, while the relationship between WUE and g_s_ was insignificant (**5b**).

**Figure 6 plants-09-01496-f006:**
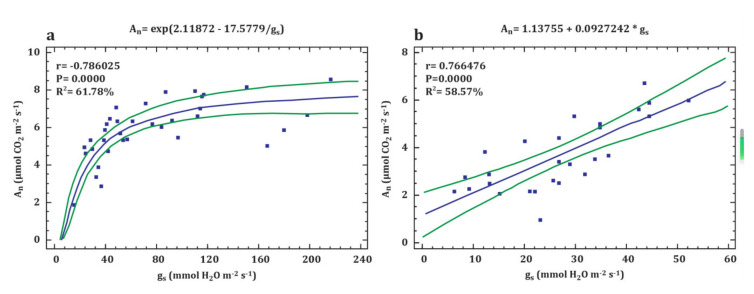
Relationship between the net photosynthetic rate (A_n_) and stomatal conductance (g_s_) of *P. pyraster* (**a**) and *S. torminalis* (**b**) seedlings grown for 50 days under different water regimes. The plot shows regression curve (blue line) and 95% confidence intervals (green lines). There was a confirmed moderately strong correlation between studied parameters for both tree species.

**Table 1 plants-09-01496-t001:** Climatic-geographic description of the original stands of woody plants [34].

Taxon	Location	Exposure	Altitude (m)	TI. (°C)	TVII. (°C)	Precipitation (mm)	Type
*P. pyraster*	Kremnica hills (Tŕnie)	S	540	−3	18	750	MW
*S. torminalis*	Central Moravian Carpathians (Vršava)	S	500	−3	18	500–550	W

TI.—the average temperature in January; TVII.—the average temperature in July; S—south exposure; MW—moderately warm region; W6—moderately warm, humid, highland climate; W—warm.

**Table 2 plants-09-01496-t002:** A two-way ANOVA analysis comparing the effects of taxon (T), drought treatment (R), and the interaction between them (T*R) on growth and biomass allocation of *P. pyraster* and *S. torminalis* seedlings in the pot experiment after 60 days of drought treatment. Significant differences (*p* < 0.05) are marked in bold. The multiple comparison of means (n = 12) was performed at the significance level of 0.05. The significant differences between species and treatments are denoted by different letters.

	*p*-Value			Control		Drought	
Parameter	T	R	T*R	*P. pyraster*	*S. torminalis*	*P. pyraster*	*S. torminalis*
Stem length (mm)	**<0.01**	**<0.01**	0.63	188.36 (±47.37) a	148.50 (±54.25) ac	117.62 (±15.11) c	88.45 (±20.40) b
Stem increment (mm)	**<0.01**	**<0.01**	0.57	161.54 (±46.09) a	122.00 (±53.21) a	86.92 (±19.00) c	59.64 (±21.48) b
Root length (mm)	0.06	**<0.01**	0.06	3494.27 (±1013.42) b	4765.23 (±1669.71) a	2861.40 (±803.77) b	2840.07(±983.51) b
Specific root length (mm·mg^−1^)	**<0.01**	**<0.01**	0.76	8.12 (±3.65) a	11.04 (±2.92) b	10.73 (±3.06) ab	14.23 (±3.76) c
Leaf area (mm^2^)	**<0.01**	**<0.01**	0.43	11,186.40 (±3571.79) a	16.515.60 (±8089.85) b	4806.37 (±1547.99) c	8083.09 (±2364.56) d
Specific leaf area (mm^2^·mg^−1^)	**<0.05**	**<0.05**	**<0.01**	19.78 (±1.11) a	19.35 (±2.22) a	16.45 (±2.49) b	19.67 (±1.85) a
Dry weight of stem (mg)	0.71	**<0.01**	0.32	500.29 (±185.16) a	536.92 (±325.40) a	287.77 (±97.57) b	209.55 (±118.07) b
Dry weight of leaves (mg)	**<0.05**	**<0.01**	0.18	567.07 (±184.30) a	886.08 (±484.80) b	299.39 (±104.10) c	412.88 (±120.87) d
Dry weight of shoot (mg)	0.14	**<0.01**	0.22	1067.36 (±356.08) a	1423.00 (±793.26) a	587.16 (±197.83) b	622.42 (±214.24) b
Dry weight of root (mg)	0.08	**<0.01**	0.72	478.64 (±171.11) a	385.75 (±215.24) ac	278.69 (±80.95) bc	217.64 (±106.81) b
Total biomass (mg)	0.48	**<0.01**	0.39	1546.00 (±503.63) a	1808.75 (±991.87) a	865.86 (±245.19) b	840.06 (±307.14) b
R:S (root to shoot ratio)	**<0.01**	0.07	0.79	0.45 (±0.10) a	0.27 (±0.07) b	0.50 (±0.18) a	0.35 (±0.09) c

**Table 3 plants-09-01496-t003:** The mean values and standard deviations (±SD) for the net assimilation rate (A_n_), transpiration rate (E), stomatal conductance to water vapor (g_s_), and water use efficiency (WUE) of the *P. pyraster* and *S. torminalis* seedlings under different water regimes (*n* = 5). The significant differences between species and treatments are denoted by different letters.

Duration of Experiment		30th Day of Treatment		40th Day of Treatment		50th Day of Treatment	
Parameter	Taxon	Drought	Control	Drought	Control	Drought	Control
g_s_ mmol H_2_O m^−2^·s^−1^	*P. pyraster*	106.00 ± 10.32 a	185.20 ± 26.47 b	49.80 ± 13.48 a	50.60 ± 20.19 a	33.60 ± 11.72 a	91.00 ± 31.21 b
	*S. torminalis*	25.8 ± 4.21 c	34.60 ± 11.85 c	21.80 ± 15.35 b	36.00 ± 11.57 ab	31.25 ± 9.74 a	26.00 ± 11.92 a
A_n_ μmol CO_2_ m^−2^·s^−1^	*P. pyraster*	6.76 ± 0.88 a	7.00 ± 1.49 a	6.44 ± 0.61 a	6.48 ± 0.94 a	4.40 ± 1.14 a	7.15 ± 1.12 b
	*S. torminalis*	2.20 ± 0.78 b	3.72 ± 1.41 b	3.88 ± 1.42 b	4.80 ± 1.76 ab	3.95 ± 1.53 a	3.45 ± 1.47 a
E mmol H_2_O m^−2^·s^−1^	*P. pyraster*	1.47 ± 0.14 b	2.44 ± 0.30 a	0.52 ± 0.13 a	0.55 ± 0.18 a	0.41 ± 0.13 a	1.03 ± 0.29 b
	*S. torminalis*	0.45 ± 0.07 c	0.60 ± 0.18 c	0.26 ± 0.17 b	0.45 ± 0.14 ab	0.40 ± 0.18 a	0.37 ± 0.15 a
WUE mmol CO_2_ mol^−1^ H_2_O	*P. pyraster*	4.60 ± 0.48 b	2.88 ± 0.57 a	13.02 ± 3.32 a	12.27 ± 1.75 a	11.42 ± 4.46 ac	7.73 ± 1.06 a
	*S. torminalis*	5.49 ± 0.29 c	6.11 ± 0.85 c	17.13 ± 5.59 a	12.93 ± 2.97 a	7.85 ± 2.47 ab	10.33 ± 1.60 bc

**Table 4 plants-09-01496-t004:** A two-way ANOVA analysis of the measured water potential of leaf tissues (Ψ_wL_) in the pot experiment with *P. pyraster* and *S. torminalis* seedlings, comparing the effects of taxon (T), drought treatment (R), and the interaction between them (T*R). Measurements were performed on 40th and 50th day of treatment (DOT). Significant differences (*p* < 0.05) are marked in bold. The multiple comparison of means was performed at the significance level of 0.05. Data are the mean values ± SD (*n* = 9). The significant differences between species and treatments are denoted by different letters.

DOT	Parameter	*p*-Value			Control		Drought	
		T	R	T*R	*P. pyraster*	*S. torminalis*	*P. pyraster*	*S. torminalis*
40th	Ψ_wl_ (MPa)	**<0.01**	0.04	**<0.01**	−0.44 (±0.10) a	−0.81 (±0.18) b	−1.27 (±0.17) c	−1.15 (±0.13) c
	RWC (%)	0.59	**<0.01**	0.05	93.62 (±3.24) a	90.59 (±1.72) ad	76.50 (±3.80) b	81.52 (±3.02) bc
50th	Ψ_wl_ (MPa)	**<0.01**	**<0.01**	0.31	−0.49 (±0.13) a	−0.84 (±0.34) b	−1.33 (±0.18) c	−1.53 (±0.08) d
	RWC (%)	0.14	**<0.01**	0.30	94.56 (±0.55) a	89.30 (±2.45) bd	79.92 (±5.83) bc	78.89 (±1.82) c

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
