# Peer review of "Physiological Performance of Pyrus pyraster L. (Burgsd.) and Sorbus torminalis (L.) Crantz Seedlings under Drought Treatment"

_plants, 2020, doi:10.3390/plants9111496_

Round 1

Reviewer 1 Report

Dear Editor,

In the current submission, the authors have addressed all the required corrections suggested. I am happy with the work that they have done. Hence, I would like to recommend the article for publication.

Author Response

Thank you for revision of  the manuscript. Your valuable comments, helped us to improve the paper.

Reviewer 2 Report

The manuscript entitled “Physiological Performance of Pyrus pyraster L. (Burgsd.) and Sorbus torminalis (L.) Crantz. Seedlings under Drought Treatment” is aimed to investigate the physiology in control and drought-stressed plants. The investigation revealed different physiological performances of the two species under both control and drought conditions. However, the hypothesis is not clearly stated, and the conclusions are merely based on physiological measurements, which makes this manuscript very descriptive, and does not contribute novel insights into the mechanistic understanding of drought adaptation. The choice of the species is not thoughtfully explained. Given the large interspecies differences observed under control conditions, the differences found under stressed conditions are not surprising and hard to explain due to the intrinsic genetic and morphological variances. The impact of this study would be much more significant if it was conducted in closely related species or species with more defined genetic background. Another major concern is the small sample size; the gas exchange was measured with only five biological replicates, which lowers the statistical power and undermines the reliability of conclusions. Additional data are needed to make the manuscript more solid. For example, repeating the experiment with more biological replicates would increase the statistical power. Biochemical and anatomical evidence would be needed to validate the conclusions drawn from the gas exchange measurements. Moreover, the writing and data presentation need substantial improvement. Therefore, I do NOT recommend this manuscript to be accepted, but I have some suggestions below to improve the work.

1. It is not clear why Pyrus pyraster L. (Burgsd.) and Sorbus torminalis (L.) Crantz. were chosen for this study. It is only mentioned in lines 86 that both species are light demanding and grow under periodic drought. If the goal is to compare the two contrasting species, then more background information on how they handle drought differently should be laid out up front in the Introduction. Some paragraphs in the Material and Methods (lines 94-106) could be incorporated into the Introduction.
2. A well-defined hypothesis is missing from the Introduction. Knowing what to expect from comparing the two species is important for experimental design, otherwise the research is very exploratory.
3. There are two Figure 1. They should be combined.
4. For all figures, the duration of drought treatment should be clearly defined.
5. Table 2. If the p-values are small than 0.01, it should be denoted as “<0.01”, rather than “0.00”. Similarly, please correct the “p = 0.0000” in line 295.
6. Figure 5 uses the subset of data from Table 3, which is redundant. Also, the positions of * and ns are confusing for readers to figure out which comparisons were made.
7. The authors stated that the plants were grown at 202.5 μmol m −2 ·s −1, why the measurements were made at 250 μmol·m −2 ·s −1?
8. How are the regression functions chosen for Figure 6 an 7. Do the regression coefficients have physiological meanings?

Reviewer 3 Report

The study presented lacks enough originality to deserve publication in Plants. It is one of the several studies with the same design, and conclusions, that have been formerly published since the previous century.

First of all, the two selected species are marginal in the overall composition of the European forest vegetation. In this sense, authors should better explain why are they chosen, further than saying that both are "light demanding woody plants which can grow even on sites with periodic drought events". The comparative studies on water stress resistance, even when done comparing only two species, are usually done after a geobotanical evidence of ecological discrimination, e.g., one species being clearly tolerant or occupying dry habitats in the nature besides other occupying wetter soils. In the case under evaluation, no geobotanical or ecological reasons are offered.

In terms of methods, authors selected stress levels based of soil water content without further explanation. Why such two levels of water deficit?Are they based on previous studies on soil water content/soil water potential curves or so on?

According to results, only the biomass allocation ratio deserves certain interest, but authors do not discuss this deeply enough. The physiological differences, with control and stressed changing throughout the whole experiment, should be better explained. 

Round 2

Reviewer 2 Report

The revised manuscript has incorporated many of my prior suggestions, which improves the writing and data presentation. Although the small sample size and lack of biochemical/anatomical supports are the major weakness of the manuscript, the results could serve as a primer for future studies. I have more suggestions for improving the revised manuscript.

  1. Table 2, “95% Tukey HSD test” is not accurate, use “at the significance level of 0.05”. Also, please specify “Mean values followed by different letters differ significantly” in terms of treatment, species, or both. Same issues for Table 4.
  2. Line 275, “documented” is confusing, use “noticed/observed” instead. Same for the “recorded” in line 286.
  3. Line 285, it seems that An was affected by the duration of the drought treatment according to figure 4b.
  4. Figure 4d, one significance letter was missed.
  5. Line 296, what do you mean by “measurements”? It would be better to specify whether it is water regime or duration of treatment.
  6. Line 413. Héroult et al. is [48] instead of [47]
  7. Please correct the “p=0.0000” in figure 5 and 6, as well as Table 4.

Reviewer 3 Report

I have read the new version of the manuscript and the new sentences and paragraphs which have been added improve the meaning of the whole paper. I still consider that the study is not very new in aim and methods, but the results are well documented and can serve as new empirical data for further meta-analysis. 

I am not able of proposing any change to improve the manuscript, as I do not consider that the writing and presentation of the data represent the principal flaw of the manuscript, but the overall aim of the study.

Authors response about the need of studying minor components of the European forest flora is convincing, and it is a good argument for publishing the paper.

Author Response

(The authors gave the same response as above.)
